# Effects of Farnesiferol B on Ischemia-Reperfusion-Induced Renal Damage, Inflammation, and NF-κB Signaling

**DOI:** 10.3390/ijms20246280

**Published:** 2019-12-12

**Authors:** Lu Zhang, Xianjun Fu, Ting Gui, Tianqi Wang, Zhenguo Wang, Gerd A. Kullak-Ublick, Zhibo Gai

**Affiliations:** 1College of Traditional Chinese Medicine; Shandong Co-innovation Center of TCM Formula; Institute for Literature and Culture of Chinese Medicine; Key Laboratory of Traditional Chinese Medicine for Classical Theory, Ministry of Education, Shandong University of Traditional Chinese Medicine, Jinan 250355, China; 2Department of Clinical Pharmacology and Toxicology, University Hospital Zurich, University of Zurich, 8006 Zurich, Switzerland; 3Mechanistic Safety, CMO & Patient Safety, Global Drug Development, Novartis Pharma, 4056 Basel, Switzerland

**Keywords:** inflammation, ischemia/reperfusion injury, Farnesiferol B, Nuclear Factor kappa-light-chain-enhancer of activated B cells (NF-κB), G-protein-coupled bile acid receptor (TGR5)

## Abstract

Background: G-protein-coupled bile acid receptor (TGR5), a membrane bile acid receptor, regulates macrophage reactivity, and attenuates inflammation in different disease models. However, the regulatory effects of TGR5 in ischemia/reperfusion (I/R)-induced kidney injury and inflammation have not yet been extensively studied. Therefore, we hypothesize that Farnesiferol B, a natural TGR5 agonist, could alleviate renal I/R injury by reducing inflammation and macrophage migration through activating TGR5. Methods: Mice were treated with Farnesiferol B before I/R or sham procedures. Renal function, pathological analysis, and inflammatory mediators were examined. In vitro, the regulatory effects of Farnesiferol B on the Nuclear Factor kappa-light-chain-enhancer of activated B cells (NF-κB) pathway in macrophages were investigated. Results: After I/R, Farnesiferol B-treated mice displayed better renal function and less tubular damage. Farnesiferol B reduced renal oxidative stress and inflammation significantly. In vitro, Farnesiferol B treatment alleviated lipopolysaccharide (LPS)-induced macrophage migration and activation, as well as LPS-induced NF-κB activation through TGR5. Conclusions: Farnesiferol B could protect kidney function from I/R-induced damage by attenuating inflammation though activating TGR5 in macrophages. Farnesiferol B might be a potent TGR5 ligand for the treatment of I/R-induced renal inflammation.

## 1. Introduction

Ischemia/reperfusion (I/R) is a common complication in patients undergoing major cardiac surgery, kidney transplantation, and those experiencing hemorrhage and dehydration [1]. Ischemia/reperfusion injury (I/RI) after renal transplantation is a well-recognized and prevalent postoperative complication, which has been thought as a risk factor for the loss of tubular epithelial cell function, leading to acute kidney injury (AKI), delayed graft function, as well as acute or chronic organ rejection [2,3]. The process of renal ischemia reperfusion (RIR) is exceedingly complex, including a series of intricate and related events that lead to renal cell injury and ultimately give rise to cell death via apoptosis and necrosis [4]. Even though reperfusion is critically paramount for the repair of ischemic tissue and the survival of patients, the additional cellular damage by itself has provoked much attention, largely due to the production of reactive oxygen species (ROS) and infiltration of neutrophils [5,6].

The mechanism and treatment of I/R on kidney injury is still under exploration. During I/R, toxic products of kidney cells and cytokine release cause damage to renal epithelial tubular cells [7]. Besides, several studies have suggested that I/R leads rise to neutrophil infiltration of kidney tissue and stimulates the oxidative stress processes producing ROS [8]. Oxidative stress directly or indirectly affects all aspects of the kidney damage by the pathway lead to apoptosis, necrosis, fibrosis, tissue damage progression, and renal dysfunction [9]. ROS is considered as both the mediator or a signaling molecule during acute kidney inflammatory diseases [10]. The oxidase activity is elevated soon after ischemic injury, which is may originate from macrophages and/or neutrophils [11]. Moreover, inflammatory cells can also induce post-hypoxic cellular damage by ROS, which in turn gives rise to tubular epithelial cell damage or death by triggering apoptosis and necrosis pathways [12,13].

Early in a classic pharmacopoeia of traditional Chinese medicine book, Ferulae Resina is recorded as widely used for its effects on removing stagnancy (Xiao Ji), dissolving hard mass (Hua Jia), dissipating obstruction (San Pi), and as having anthelmintic properties (Sha Chong). Previous research reported the isolation of Farnesiferol B, a new sesquiterpene that is highly produced in the Apiaceae family, from the roots of Ferula [14,15]. Farnesiferol B has shown anti-plasmodial activity and anti-cytotoxicity effects in rat skeletal fibroblasts, a L6-cell line [16]. In a recent in silico work, Farnesiferol B was confirmed as a TGR5 agonist, also known as a G-protein-coupled bile acid receptor 1 (GPBAR1) activator, which denotes a regulatory effect of Farnesiferol B on inflammation [17].

TGR5 is expressed in several types of cells, including hepatic non-parenchymal cells [18,19,20]. The renal tubular cells, and immune cells including monocytes and macrophages [21,22]. Recently, it was demonstrated that TGR5 activation could be exploited to confer protection from diabetic nephropathy [23]. Considering that TGR5 activation possesses anti-oxidant and anti-inflammatory roles [23,24] in the kidney, we hypothesized that Farnesiferol B would alleviate renal I/R injury by reducing inflammation and ROS through regulating renal tubular cells and macrophages over TGR5.

In this study, the protective effect of Farnesiferol B was examined in a mouse model of AKI induced by I/R. We investigated the regulatory effects of Farnesiferol B on pro-inflammatory pathways in LPS-treated macrophages.

## 2. Results

### 2.1. Farnesiferol B Protection for Kidney Damage in I/R Kidney

The overall effect of Farnesiferol B on kidney injury was first explored by histological assessment and creatinine quantification. Haemotoxylin and Eosin (HE) staining and kidney injury marker Kim-1 (kidney injury molecule-1) immunohistochemical staining analysis results show that I/R kidneys are characterized by tubular basal membrane rupture, nuclear infiltration, tubular vacuolization, higher Kim-1 expression, and a higher renal tubular injury score (Figure 1A,B,E). Levels of proximal tubular cell death and cell proliferation were assessed. In the TUNEL assay, the nuclei of TUNEL-positive cells were stained red (Figure 1C), and the levels of cell death were indicated as the percentage of TUNEL-positive cells (Figure 1F). TUNEL-positive cells were observed mainly in the tubular area of the renal cortex. I/R group displayed more TUNEL-positive cells that the sham group (Figure 1C(b) vs. (a)). PCNA expression was observed in proliferating cell nuclei, and was significantly increased in AKI compared with the sham group (Figure 1D,G). Serum creatinine levels in I/R mice were significantly elevated in comparison to sham-operated mice (Figure 1H). The treatment with Farnesiferol B prior to I/R decreased serum creatinine level, attenuating the I/R-induced kidney injury (Figure 1A,C,H). However, Farnesiferol B treatment did not change the I/R-induced cell proliferation (Figure 1D(c) vs. (b)). The above results suggest that, in general, Farnesiferol B treatment protects against the kidney damage caused by I/R.

### 2.2. Farnesiferol B Reduces Oxidative Stress and Lipid Oxidative Signaling Pathways in I/R Kidney

I/R is often associated with oxidative stress, with existing evidence suggesting that oxidative stress is a paramount contributor in causing kidney damage [25]. Therefore, the effect of Farnesiferol B on oxidative stress in the kidney after I/R was evaluated. Immunohistochemical staining for NGAL neutrophil gelatinase-associated lipocalin (NGAL), an oxidative stress risk factor, showed that I/R may induced significant increases of oxidative stress in I/R kidneys (Figure 2A(b),B). The level of NGAL was reduced in mice treated with Farnesiferol B (Figure 2A(c),B). The potential mechanisms involved in the inhibitory effects of Farnesiferol B on I/R-induced oxidative stress were investigated. The I/R greatly increased the oxidative stress production and impaired antioxidant capacity in the injured kidney (Figure 2C–E). Farnesiferol B administration significantly diminished oxidative stress in the urine of injured group, namely H_2_O_2_ (hydrogen peroxide). Treatment with Farnesiferol B significantly increased the expression of Nrf2 and its downstream HO-1 (Figure 2D,E).

Reactive oxygen species accumulation can lead to lipid peroxidation and ferroptosis, a kind of regulated cell death [26]. The lipid peroxidation marker, 4-HNE (4-hydroxynonenal) and MDA (malondialdehyde), and markers related to ferroptosis were examined. The results show that 4-HNE and MDA levels were induced and GSH (glutathione) levels were reduced in the injured kidney (Figure 2F–H). Farnesiferol B administration reduced kidney lipid peroxidation and induced GSH level in the renal tissue homogenate. Furthermore, mRNA expression of Gpx4, the key ferroptosis regulator, was examined (Figure 2I). Gpx4 mRNA level was significantly down-regulated after I/R injury, whereas the expression seemed to be increased by Farnesiferol B treatment (not significantly). Taken together, the results indicated that anti-lipid peroxidation effects seen in Farnesiferol B treatment group could be an indirect result of its regulation on antioxidant pathways.

### 2.3. Farnesiferol B Protectes Kidney from I/R-Induced Inflammation and Inhibits NF-κB Signaling Pathway

The other oxidative stress-producer is the inflammatory cell such as monocytes and macrophages, which infiltrate into tissue, especially during acute inflammation [27]. Next, we analyzed the degree of kidney inflammation. I/R increased the positive stainings of macrophages and neutrophils in the kidney (Figure 3A–D). I/R also induced levels of TNFα and MCP-1 in mouse serum and kidney, as well as the proinflammatory mediator LTB_4_ in the kidney (Figure 3E–I). The number of macrophages and neutrophils, as well as the serum and kidney levels of TNFα, MCP-1, and LTB4 were significantly reduced by Farnesiferol B treatment (Figure 3E–I). Inflammation-related genes expressed in the kidney were also measured. Quantitative analysis showed that I/R stimulated the expression of kidney TNFα, IL-6, and Icam mRNA levels, while levels of these mRNA decreased under treatment with Farnesiferol B (Figure 3J–L). The evidence presented here suggests a positive role of Farnesiferol B in attenuating renal inflammation.

NF-κB signaling pathway is one of the key inflammatory pathways during AKI. To examine the inhibitory effects of Farnesiferol B on I/R-induced inflammation, p65 levels from different treatment groups were evaluated. I/R induced increased tubulointerstitial staining of p65 in the kidney sections (Figure 4A(b)) and p65 translocation in the nuclei (Figure 4B). The I/R+Farnesiferol B group showed reduced levels of p65 immunostaining in kidney sections as well as nuclei translocation, indicating that part of anti-inflammatory effect is reacted on inflammatory cells.

### 2.4. Farnesiferol B Inhibits the Activity of Inflammatory Cells by Activating TGR5

To examine the direct effect of Farnesiferol B on macrophages, we conducted in vitro experiments using the J774 cell line. J774 macrophages were treated with LPS and/or Farnesiferol B and migration ability in a transwell system was monitored as a proxy for infiltration capability. Representative images show that migration induced by LPS was abolished by co-incubation with Farnesiferol B (Figure 5A). LPS-induced migration was associated with higher mRNA expression levels of TNFα, Ccl2, Ccl3 in macrophages, compared with that from the untreated cells. Treatment with Farnesiferol B lowered mRNA expression of these inflammatory cytokines significantly (Figure 5B–D).

Farnesiferol B was previously reported to be a TGR5 agonist, which can inhibit macrophage infiltration through inhibiting NF-κB activation [28]. Based on this, we speculated that the anti-inflammatory effect of Farnesiferol B may be related to its ability to activate TGR5. Therefore, we evaluated the regulatory effect of Farnesiferol B on LPS-induced NF-κB activation in J774 macrophages. Immunoblotting of nuclear protein for NF-κB p65 indicated an LPS-induced nuclear translocation in J774 macrophages (Figure 5E). Similar to INT777, a known TGR5 agonist, Farnesiferol B also significantly reduced p65 translocation in a dose-dependent way (Figure 5E). PARP-1 was stably expressed between groups as a nucleoprotein control (Figure 5E). An NF-κB binding assay also revealed that LPS-induced NF-κB binding activity was blocked by Farnesiferol B or INT777 co-incubation (Figure 5F), which was consistent with previous publications [29].

In order to further verify the target of Farnesiferol B in macrophages, RAW264.7 cells, which do not express TGR5 [30], were exposed to LPS with and without Farnesiferol B. As shown in Figure 6, Farnesiferol B could not inhibit LPS-induced cell migration, suggesting that Farnesiferol B suppresses LPS-mediated macrophage migration by activating TGR5. Benzoxathiole derivative (BOT), an NF-κB inhibitor, was used as a positive control to block LPS-induced migration (Figure 6c). Taken together, Farnesiferol B inhibits the LPS-induced NF-κB pathway by activating TGR5.

## 3. Discussion

In the current study, our results point out that Farnesiferol B treatment play a beneficial role in renal I/R injury. Compared with I/R treated mice, Farnesiferol B-treated mice showed lower score of renal injury, while significantly decreased levels of histological tubular injury, oxidative stress, and inflammation (Figure 7). In line with and based on our experiments in mice, the in vitro analysis show that Farnesiferol B reduces LPS-induced macrophage migration. Moreover, Farnesiferol B can inhibit NF-κB nuclear translocation through activating TGR5 in macrophages. Thus, Farnesiferol B might represent a novel natural compound against I/R-induced kidney damage.

Ischaemia-reperfusion (I/R) injury is the main cause of AKI under common clinical conditions [31,32]. Nowadays, the pathogenesis of AKI is characterized by renal tubular damage, inflammation, and vascular dysfunction [33,34]. In vivo experiment, we examined the kidney of AKI mice and found that Farnesiferol B effectively reduced the regulated cell death and oxidative stress. Although it was hard to identify the exact type of cell death, both results of TUNEL assay and ferroptosis marker Gxp4 showed that Farnesiferol B have protective effects on tubular cell damage during AKI. NGAL is an early biomarker of AKI which is produced in the distal nephron and its synthesis is upregulated in response to kidney injury [35]. Recent evidence demonstrates that high NGAL level is a risk factor for oxidative stress in patients [36]. In our case, I/R increased highly expression of NGAL in mice. Furthermore, Farnesiferol B reduced the NGAL level both in kidney and in urine, as well as the other oxidative stress marker MDA. Furthermore, with treatment of Farnesiferol B, GSH levels was increased and Nrf2 and HO-1 expression were restored in the I/R injured kidney, while at the same time, reduced oxidative stress (H_2_O_2_) and lipid peroxidation (4-HNE and MDA). These data indicated that antioxidant related pathways may be regulated by Farnesiferol B-TGR5 signaling pathway. Nrf2 and its target gene, HO-1, has generally been considered to be an adaptive cellular response to oxidative stress [26]. Recently, Nrf2/HO-1 has also been shown to be protective in AKI and diabetic nephropathy [37,38,39]. It has been reported that treatment with either FXR/TGR5 dual agonist INT-767 or TGR5 specific agonist INT-777 could prevent diabetic nephropathy through inducing Nrf2-mediated antioxidant generation, and reducing renal expression of oxidative stress [23,40]. We have seen similar effects in the I/R-induced mouse AKI model with treatment with FXR/TGR5 dual agonist [25]. In the study, treatment with 6alpha-ethyl-chenodeoxycholic acid (6-ECDCA), a potent dual FXR/TGR5 agonist [41], significantly improved Nrf2-mediated antioxidant capacity. Silencing of Nrf2 blocked the antioxidant effect of 6-ECDCA in proximal tubule cells exposed to hypoxia. Based on these findings, TGR5 could be a potent target to induce antioxidant pathways. However, to identify the renal signaling pathways regulated by FXR and TGR5 respectively and specify the regulatory effects of TGR5 on antioxidant pathways need to be further studied.

Renal inflammation after I/R is directly related to monocyte infiltration and macrophage activation [25,42]. TGR5, also called GPBAR1 or GPR131, is a bile acid–responsive G protein-coupled receptor, which plays a crucial role in protection against diet-induced diabetes through different cellular mechanisms [43]. The role of TGF5 activation in modulating inflammatory pathways was confirmed in experiments on mice and immune cells [21,28]. Recent evidence suggests that activation of TGR5 regulates inflammatory cell signaling pathways such as NF-κB, AKT and extracellular signal-regulated kinase (ERK) [44,45]. Macrophages play a pivotal role in kidney injury, inflammation, and fibrosis [46]. In the present study, we can clearly notice that, after I/R, the mice shown an increased inflammation level on renal tissue (e.g., the infiltration of macrophages and neutrophils in Figure 3 and high expression of p65 in Figure 5), as well as high level of proinflammatory factors (TNFα, MCP-1, IL-6) both in serum and kidney tissue. While the treatment with Farnesiferol B demonstrated a critical anti-inflammatory effects by reducing those inflammatory factors or NF-κB activation. However, NF-κB signaling can be activated in all cell types during AKI, thereby immunostaining or immunoblotting providing only limited information about cell-specific NF-κB functions in the kidney during AKI [47]. Therefore, we employed the in vitro experiments. Our results shown that Farnesiferol B effectively inhibited the expression of inflammatory cytokines, such as MCP-1, LTB4, and TNFα. Farnesiferol B inhibited LPS-induced NF-κB activation and decreased p65 translocation in J774 macrophages, possessing similar effects as other TGR5 agonists [40]. Moreover, the anti-inflammatory effect of Farnesiferol B was blocked in RAW264.7 cells, which do not express TGR5 [21,30]. Taken together, these results indicate that Farnesiferol B inhibits LPS-induced NF-κB activation through activating TGR5, which is consistent with previous predictive results [17].

Ferula species from the family Apiaceae are rich sources of biologically active natural products including sesquiterpene coumarins (SCs) and sesquiterpenes [48]. Some studies have shown that SCs are able to enhance the cytotoxicity of anticancer compounds [49]. Recently Kasaian et al. showed enhancement of doxorubicin cytotoxicity in MCF-7/Adr cells (doxorubicin resistant derivatives of MCF-7 cells overexpressing P-gp) when combined with non-toxic concentrations of farnesiferols, proving the significant activity of Farnesiferol B on multi-drug resistant cells [50]. Previous studies show that sesquiterpene coumarins and their derivative from *Ferula fukanensis* reduces IL-6 and TNFα, while inducing nitric oxide synthase in RAW264.7 cells, possessing inhibitory effects on LPS and IFN-γ induced pro-inflammatory cytokine release and nitric oxide production [51,52]. A recent study also showed the anti-inflammatory effect of *Ferula szowitsiana* in vitro, which indicates that Farnesiferol B might inhibit neuroinflammation via reducing the generation of inflammatory cytokines [53]. In this study, we show the active effect of Farnesiferol B on TGR5, which suppresses the NF-κB p65 binding activity and inhibits macrophage migration.

## 4. Materials and Methods

### 4.1. Animals Study Approval and Tissue Samples

All animal experiments conformed to both Swiss and Chinese animal protection laws and were approved (May, 2015) by the Scientific Animal Study Committee of Shandong University, Jinan, China (study number 2015064).

Six-week-old female C57/BJ mice were randomly assigned to I/R or sham procedures. They were divided into four groups with six animals each: sham, I/R, I/R + Farnesiferol B and sham + Farnesiferol B. AKI was induced by unilateral nephrectomy and contralateral ischemia and reperfusion, as previously described [25]. For Farnesiferol B treatment, mice were injected intraperitoneally (i.p.) with Farnesiferol B (10 mg/kg, Golexir Pars Co., Mashhad, Iran) 2 h before the procedure. All mice were killed under anesthesia 24 h after surgery. Kidneys were harvested for further analysis.

### 4.2. Measurements in Serum, Urine, and Kidney Samples

Urinary H_2_O_2_ and NGAL levels were measured in the resulting urine samples with the Amplex Red H_2_O_2_ assay kit (A12214, Invitrogen, Carlsbad, CA, USA) and Mouse Lipocalin-2/NGAL ELISA Kit (ab119601, Abcam, Cambridge, UK). Serum creatinine, MCP-1, TNFα, and LTB4 levels were measured with a creatinine assay kit (ab65340, Abcam, Cambridge, UK), Mouse MCP1 ELISA Kit (ab100721, Abcam), Mouse TNF alpha ELISA Kit (ab46105, Abcam), and LTB4 Parameter Assay Kit (KGE006B; R&D Systems, Minneapolis, MN, USA), respectively. Kidney malondialdehyde (MDA) levels were measured by a Lipid Peroxidation (MDA) Assay Kit (ab118970, Abcam).

### 4.3. Renal Pathological Assessments and Immunostaining

Tissue sections were stained with hematoxylin and eosin (HE) using standard protocols. TUNEL staining was performed with an ApopTag kit (Millipore, Billerica, MA, USA) based on the manufacturer’s instructions. The antibodies for immunohistochemistry used in this study were those against Kim-1 (ab78494, Abcam), 4-hydroxynonenal (4-HNE, ab46545, Abcam), neutrophil gelatinase-associated lipocalin (NGAL, ab63929, Abcam), Ly-6B (NBP2-13077), NFκB p65 (sc-372, Santa Cruz, CA, USA), and MAC387 (ab22506, Abcam). Sections were treated with the Envision+ DAB kit (Dako, Basel, Switzerland) according to the manufacturer’s instructions.

### 4.4. Cell Culture and Migration Assay

J774 cells were grown in Dulbecco’s modified Eagle’s medium and RAW264.7 macrophages were maintained in RPMI 1640 medium. Both cell culture media were supplemented with 10% FCS, 100 U/mL penicillin, and 100 mg/mL streptomycin. Cells were cultured at 37 °C in a humidified atmosphere with 5% CO_2_.

For LPS treatment, J774 or RAW cells were treated with 100 ng/mL LPS with or without co-incubation of 20 μM Farnesiferol B for 2 h. Afterwards, RNA and protein were extracted for further analysis. For the migration assay, cells were seeded in 3-µm pore polycarbonate membrane inserts (Costar Corning, Darmstadt, Germany) on 12-well plates and treated with 100 ng/mL LPS (L3254, Sigma, St. Louis, MO, USA) or 20 μM Farnesiferol for 2 h at 37 °C. The inserts were washed, fixed, and stained with crystal violet for analysis.

### 4.5. NF-κB DNA Binding Assay

The assay was performed with a NF-κB p65 Transcription Factor Assay kit (ab133112, Abcam). Briefly, nuclear extracts from cells were incubated with a double stranded DNA sequence containing the NF-κB response element overnight. After washing, an NF-κB antibody was added and incubated for 1 h. Then an HRP-conjugated secondary antibody was added for 1 h and washed twice. After incubation with developing solution for 30 min, stop solution was added and the results were analyzed with a microplate reader (Glomax, Promega, Madison, WI, USA).

### 4.6. Isolation of RNA from Kidney Tissue and Quantitative Real-Time Polymerase Chain Reaction (qRT-PCR)

Total RNA was prepared using Trizol (Invitrogen, Carlsbad, CA, USA). The mRNA was quantified based on absorbance at 260 nm. After DNAse (Promega) treatment, 2 μg total RNA was reverse transcribed using oligo-dT priming and Superscript II (Invitrogen). First-strand complementary DNA was used as the template for real-time polymerase chain reaction analysis with TaqMan master mix and primers (Life Technologies, Carlsbad, CA, USA). Primers used were TNFα (No. Mm00443258-m1), Icam1 (No. Mm00516023_m1), IL-6 (No. Mm00446190_m1), Ccl2 (No. Mm00441242_m1), Ccl3 (No. Mm00441259_g1), Gpx4 (No. Mm00515041_m1), Nrf2 (No. Mm00477784_m1), HO-1 (No. Mm00516005_m1). Transcript levels, determined in two independent complementary DNA preparations, were calculated and expressed relative to levels of RNA for the housekeeping gene beta-actin (No. Mm00607939-s1) or GAPDH (No. Mm99999915-g1).

### 4.7. Western Blotting

Protein lysates (20 μg protein) from cells nuclear were separated by SDS-PAGE and blotted on polyvinylidene difluoride membranes (Millipore, Burlingtob, MA, USA). The membranes were incubated overnight at 4 °C with the respective primary antibodies and secondary antibodies accordingly. Staining was then developed using the ECL Plus detection system (Amersham Biosciences, Little Chalfont, UK). The antibodies used were anti-p65 (sc-372, Santa Cruz, Recife, Pernambuco) and PARP-1 (AV33754, Sigma-Aldrich).

### 4.8. Statistics

Data are expressed as means ± SD. For data relating to baseline characteristics and histological analysis, groups were compared by one-way ANOVA followed by Bonferroni’s test. Statistical analyses were performed using GraphPad software (GraphPad Software Inc., San Diego, CA, USA).

## 5. Conclusions

In vivo, the present study clearly demonstrates that Farnesiferol B protects kidney from I/R-induced damage by reducing oxidative stress and inflammation. In vitro, Farnesiferol B ameliorates macrophage migration by activating TGR5.

## Figures and Tables

**Figure 1 ijms-20-06280-f001:**
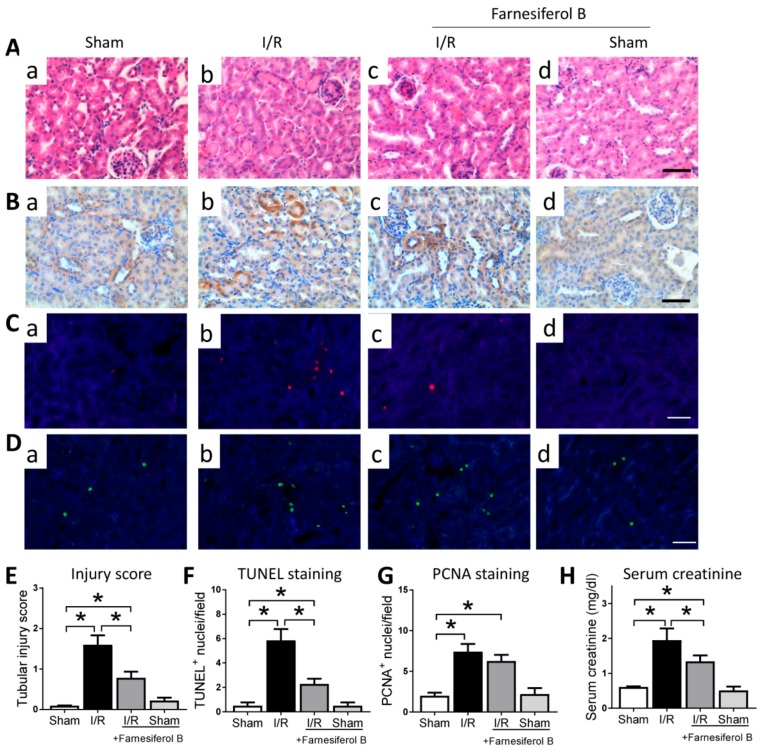
Farnesiferol B protects kidney from I/R-induced kidney damage. Representative images showing (**A**) Haemotoxylin and Eosin (HE) staining and (**B**) Kim1, (**C**) TUNEL, (**D**) PCNA immunostaining on renal sections from (**a**) sham, (**b**) I/R, (**c**) I/R + Farnesiferol B and (**d**) sham + Farnesiferol B groups (scale bar 50 μm). Quantitative analysis of (**E**) tubular injury scores, (**F**) TUNEL staining, (**G**) PCNA staining, and (**H**) serum creatinine from different groups. *n* = 6 mice/group. Data are means ± SD, one-way ANOVA with Bonferroni’s test. * *p* < 0.05.

**Figure 2 ijms-20-06280-f002:**
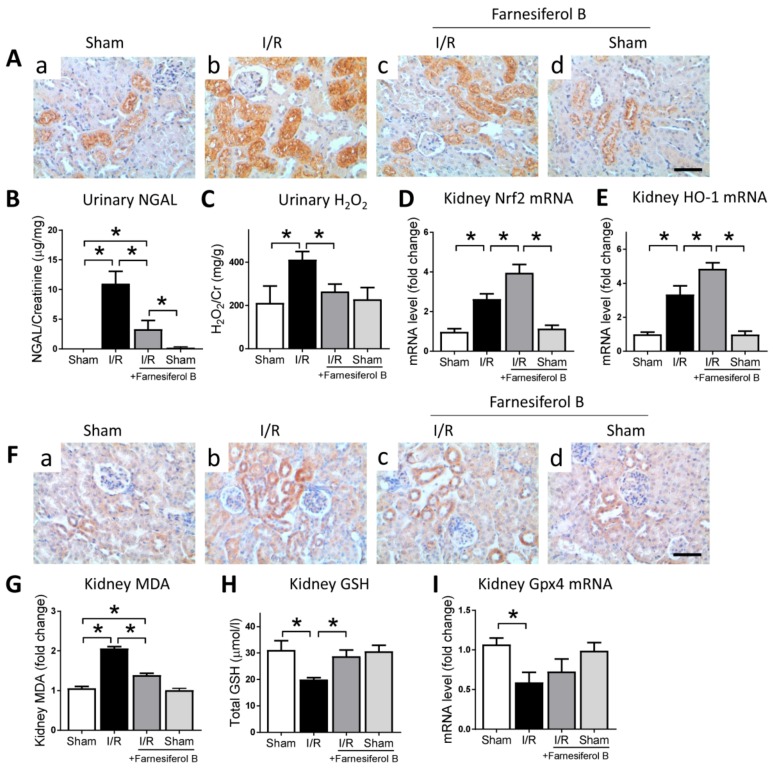
Farnesiferol B reduces oxidative stress in ischemia/reperfusion (I/R) kidney. (**A**) Representative images of immunostaining for NGAL on renal sections from (**a**) sham, (**b**) I/R, (**c**) I/R + Farnesiferol B  and (**d**) sham +  Farnesiferol B groups (scale bar 50 μm). (**B**) urinary neutrophil gelatinase-associated lipocalin (NGAL), (**C**) urinary H_2_O_2_, and kidney mRNA levels of (**D**) Nrf2 and (**E**) HO-1 were analyzed. *n* = 6 mice/group. Data are means ± SD, one-way ANOVA with Bonferroni’s test. * *p* < 0.05. (**F**) Representative images of immunostaining for 4-HNE on renal sections from (**a**) sham, (**b**) I/R, (**c**) I/R + Farnesiferol B  and (**d**) sham + Farnesiferol B groups (scale bar 50 μm). (**G**) kidney malondialdehyde (MDA), (**H**) kidney GSH, and kidney mRNA levels of (**I**) Gpx4 were analyzed. *n* = 6 mice/group. Data are means ± SD, one-way ANOVA with Bonferroni’s test. * *p* < 0.05.

**Figure 3 ijms-20-06280-f003:**
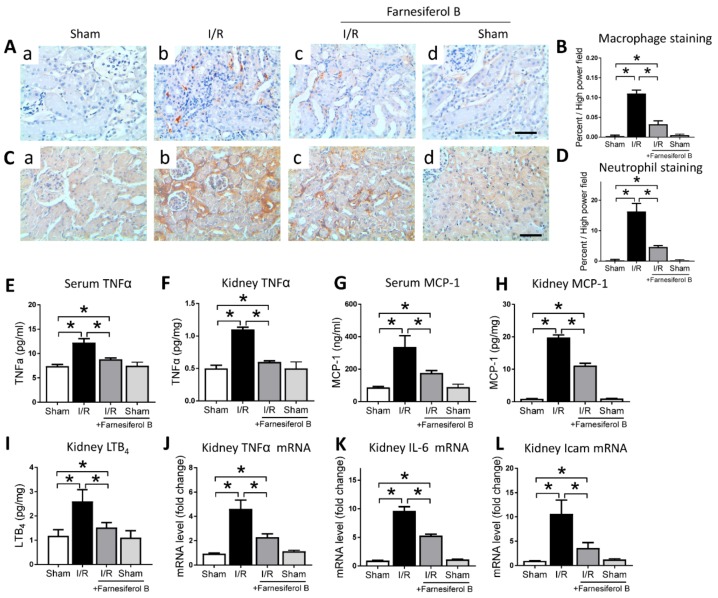
Farnesiferol B protects kidney from I/R-induced inflammation. (**A**) Representative images and (**B**) the quantitative analysis of positive immunostaining for macrophages on renal sections from (**a**) sham, (**b**) I/R, (**c**) I/R + Farnesiferol B, and (**d**) sham + Farnesiferol B groups (scale bar 50 μm). (**C**) Representative images and (**D**) the quantitative analysis of positive immunostaining for neutrophils on renal sections from (**a**) sham, (**b**) I/R, (**c**) I/R + Farnesiferol B,  and (**d**) sham + Farnesiferol B groups (scale bar 50 μm). Levels of (**E**) serum TNFα, (**F**) kidney TNFα, (**G**) serum MCP-1, (**H**) kidney MCP-1, (**I**) kidney LTB_4_, (**J**) kidney TNFα mRNA, (**K**) kidney IL-6 mRNA, and (**L**) kidney Icam mRNA from different treatment groups. *n* = 6 mice/group. Data are means ± SD, one-way ANOVA with Bonferroni’s test. * *p* < 0.05.

**Figure 4 ijms-20-06280-f004:**
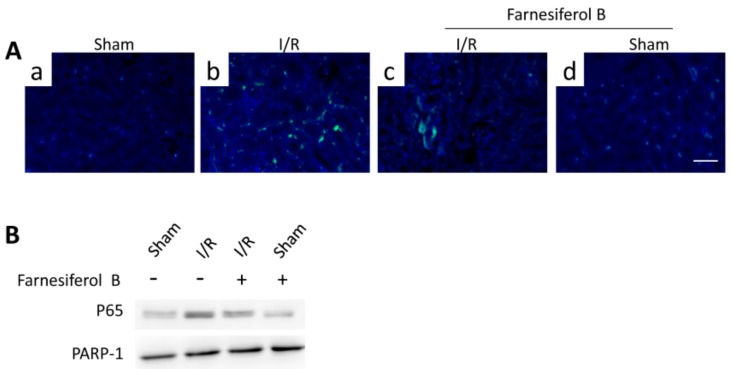
Farnesiferol B inhibits NF-κB signaling pathway in I/R kidney. (**A**) Representative images showing immunostaining of p65 on renal sections from (**a**) sham, (**b**) I/R, (**c**) I/R + Farnesiferol B and (**d**) sham + Farnesiferol B groups (scale bar 50 μm). (**B**) Representative immunoblotting image of nuclear protein for NF-κB p65 expression in kidney tissue. PARP-1 protein is used as the loading control.

**Figure 5 ijms-20-06280-f005:**
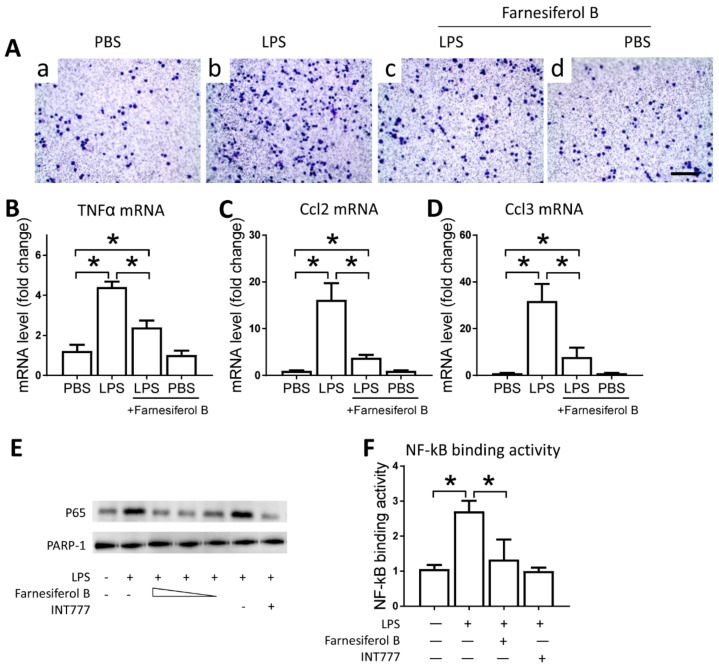
Farnesiferol B inhibits the activity of inflammatory cells by activating TGR5. (**A**) Representative images of J774 cells seeded in 3-µm pore polycarbonate membrane and stained with crystal violet for migration analysis (scale bar 100 μm) after treatment with (a) PBS, (b) LPS, (c) LPS+Farnesiferol B, or (d) PBS+Farnesiferol B. Level of mRNA for (**B**) TNFα, (**C**) Ccl2 and (**D**) Ccl3 with different treatments. (**E**) Representative immunoblotting image of nuclear protein for NF-κB p65 expression. J774 cells were treated with 100 ng/mL LPS with or without co-incubation with Farnesiferol B at 5 uM, 10 uM and 20 µM, or INT777 for 2 h. (**F**) NF-κB DNA binding activity. Data are means ± SD of at least three independent experiments, one-way ANOVA with Bonferroni’s test. * *p* < 0.05.

**Figure 6 ijms-20-06280-f006:**
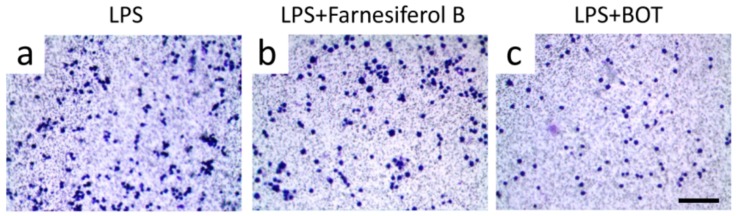
Farnesiferol does not inhibit migration of inflammatory cells not expressing TGR5. Representative images of RAW 264.7 macrophages seeded in 3-µm pore polycarbonate membrane with different treatment for 2 h, and stained with crystal violet for migration assay (scale bar 100 μm). Cells were treated with (**a**) 100 ng/mL LPS, (**b**) LPS with 20 µM Farnesiferol co-incubation, and (**c**) LPS with a NF-κB inhibitor, benzoxathiole derivative (BOT).

**Figure 7 ijms-20-06280-f007:**
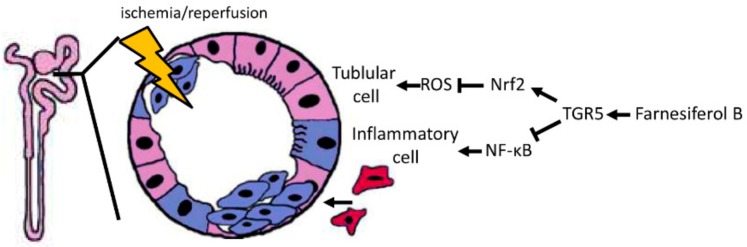
Model of the Farnesiferol B-mediated protection against ischemia/reperfusion-induced AKI. I/R induces ROS generation that causes tubular cell damage (shown in purple-blue). In parallel, inflammatory cell infiltration (shown in red) also promotes oxidative stress and post-hypoxic kidney damage after I/R. Farnesiferol B-mediated TGR5 activation induces the antioxidant Nrf2 pathway and inhibits the proinflammatory NF-κB signaling pathway, which in turn protects the kidney from tubular damage and inflammation.

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
