# Peer review of "Effects of Farnesiferol B on Ischemia-Reperfusion-Induced Renal Damage, Inflammation, and NF-κB Signaling"

_ijms, 2019, doi:10.3390/ijms20246280_

Round 1
Reviewer 1 Report
Manuscript ID: ijms-644255
Type of manuscript: Article
Title: Effects of Farnesiferol B on Ischemia-Reperfusion-induced Renal Damage, Inflammation, and NF- κB Signaling
Authors: Lu Zhang , Xianjun Fu , Ting Gui , Tianqi Wang , Zhenguo Wang , Gerd A. Kullak-Ublick * , Zhibo Gai *
In this original article entitled “Effects of Farnesiferol B on Ischemia-Reperfusion-induced Renal Damage, Inflammation, and NF- κB Signaling” Lu and colleagues studied the role of Farnesiferol B in renal damage induce by I/R. The authors delivered an interesting manuscript dealing with Farnesiferol B, a natural TGR5 agonist and Renal inflammation and acute kidney injury, which is certainly interesting for the readership of Journal of molecular sciences.
This is a highly interesting, but for publication a few points should be re-evaluated and would improve the manuscript.
1)The acute kidney injury (AKI) is characterized by an increase of cell death via apoptosis and regulated necrosis. The authors described that Farnesiferol B modulates apoptosis assessed by TUNEL staining. This affirmation in not correct because TUNEL staining only gives information about the generic cell death but is not a specific marker of apoptosis; for that the authors need to check other specific indicators of apoptosis such as caspase 3. Is necessary to correct this in the text and indicates that the modulation is not a modulation of apoptosis, is a modulation of cell death.
2) The author described that NGAL is an oxidative stress marker but this is nor correct. NGAL is a marker of renal damage and I have not found scientific information that corroborate the author’s affirmation. On the other hand, There are studies that determine as higher level of NGAL is a risk factor for oxidative stress but this is not the same idea that the authors showed in their manuscript. Is necessary to correct this mistake. For detecting ROS there are several markers such as the molecular probe 2′,7′-dichlorodihydrofluorescein diacetate (H2DCFDA) ; dihydroethidium (DHE) for detection of superoxide anion production and also MitoSOX Red to detect mitochondrial superoxide production. All of them are possibilities to detect ROS production to complete the study.
3) In the line 109 the authors described that the treatment with Farnesiferol B significantly increased the expression of Nrf2 and its downstream gene HO-1. The increase of the antioxidant response is representative of an increase of damage and oxidative stress. Because of this, If the treatment with Farnesiferol B induces an increase of NRF2 and HO-1, this situation indicates more damage and more oxidative stress. This result is opposite to the vast majority of the results and the principal idea that the authors want to transmit in the manuscript because the treatment with farnesiferol B reduces renal damage (inflammation, renal function etc…) . How do you explain this?
4) In relation with NFkB signalling, the authors showed in figure 4 a p65 staining in the interstitial space in I/R mice but the western blot of renal tissue extracts showed an increase of nuclear p65. In total renal extracts you have all the cells population of renal cells with an elevated percentage of tubular cells. moreover, previous studies described that after I/R kidney injury there is a p65 activation in tubular cells (J Am Soc Nephrol 27: 2658–2669, 2016). How do you explain that you do not have any staining in tubular cells?
5) In the line 110, In relation with the Ferroptosis is necessary to introduce more information to clarify the type of necrosis regulated and the reason for that the authors decided to analyse ( references). In this lines there is an error of concept, ferroptosis is not a type of apoptosis, is another type of cell death; this is necessary to clarify for the lector. What happen with the other indicator of the ferroptosis activation IL-33?
Reviewer 2 Report
Review Report
Brief summary
The paper reports the anti-inflammatory and anti-oxidant effects of the herbal compound Farnesiferol B in vivo and in vitro. Its shows the pathway via stimulation of the receptor TGR5. Focus are the effects of Farnesiferol B on kidney damages. The experiments with macrophages extend this view to a more general principle.
Broad comments
The paper is sound and round. A complete set of experiment were done on mRNA, protein expression level and tissue and more. All relevant controls were included, the effects are significant. The results of this study are new and well defined. All results are interpreted appropriately; arguments are supported by correct citations. The authors do not tend to speculate at all, even though there would be space enough for further going speculations.
The language of the paper is clear and precise. Figures and caption are clear and meaningful. The quality of presentation is on the highest level and paper writing routine of the authors is obvious. The study is correctly designed and technically sound. All conclusions base on robust data. Methods, tools, software, and reagents are described with sufficient details to allow another researcher to reproduce the results.
The topic is not only relevant for experts in the field of kidney diseases; Farnesiferol B most probably will be effective in many other diseases, too. The report is full in the scope of the International Journal of Molecular Sciences. The authors extend existing knowledge by contributing these new results.
Specific comments
The manuscript is provided as a pdf file. It is a little annoying that 30% of the space is occupied by an almost empty field, with few edits from the past. All comments and track changes should be removed in a submitted manuscript, to make reading easier.
Figure 3 K, IL-6 is written as Il6, what is unusual. Better use upper case L, usual are IL-6 and IL6. Later in the text the authors use IL-6.
Figure 3 and Figure 5 Inconstant writing for TNFα in Figure 3 with the Greek letter alpha, in Figure 5 with a, instead of alpha.
Line 47/48 the citation [8] “Basile, D. P., The endothelial cell in ischemic acute kidney injury...2007” does neither seem to support the argument of neutrophil infiltration, nor oxidative stress producing ROS. Do you mean citation [11]?
Line 199 It must read Figure 6 instead of Figure 5.
Line 230 update citation: …stress [26](PMID:25268649).
Line 252 missing blank (ERK)[38..
Line 253 missing blank fibrosis[40]
Line 259 missing blank results[17]
Line 267 “andTNFα” (missing blank)
Line 307 Please specify which LPS was used (supplier and order number). There are serious differences between the different products (TLR4, TLR2 etc.)
Line 310 uM instead of µM
Line 325 missing blank
Line 329 missing blank
Line 342-354 Conclusion
To my taste, the conclusion is okay, but a little short and does not fully reflect the many results, e.g. in vivo/in vitro.
Suggestions for discussion, but not requirements
Since nuclear translocation of NF-κB and the generation of oxidative stress are fundamental processes of inflammation, Farnesiferol B most probably can be helpful for treatment of many diseases. The authors do not speculate, but to my opinion, they might encourage other researchers to test Farnesiferol B in other disease models.
Many other natural compounds with anti-inflammatory and anti-oxidant properties where described during the last years. Do the authors see any realistic chance of translation of Farnesiferol B into clinical routine?
Did the authors observe any negative side effects like reduced viability in cell culture?
Overall Recommendation
Accept after Minor Revisions.
Round 2
Reviewer 1 Report
There is no new comments, the manuscript is accepted in present form
This manuscript is a resubmission of an earlier submission. The following is a list of the peer review reports and author responses from that submission.
Round 1
Reviewer 1 Report
In this original article entitled “Effects of Farnesiferol B on Ischemia-Reperfusion-induced Renal Damage, Inflammation, and NF- κB Signaling” Lu and colleagues studied the role of Farnesiferol B in renal damage induce by I/R. The authors delivered an interesting manuscript dealing with Farnesiferol B, a natural TGR5 agonist and Renal inflammation and acute kidney injury, which is certainly interesting for the readership of Journal of molecular sciences.
This is a highly interesting and well-written manuscript, but for publication a few points should be re-evaluated and would improve the manuscript.
MAJOR POINTS
The acute kidney injury (AKI) is characterized by an increase of cell death via apoptosis and regulated necrosis, but the authors do not include any studies about the role/effect of Farnesiferol B in the cell death a key step in the acute renal damage.Is necessary to add new studies about the role of Farnesiferol B in the tubular cell death in the kidney during I/R. The authors can include studies such as TUNEL, or the analysis of caspase 3 and caspase 8 (components of the apoptosis pathway). Martin-Sanchez et al (Proc Natl Acad Sci U S A. 2018 Apr 17;115(16):4182-4187) previously observed that the initial wave of cell death in AF induced AKI was ferroptosis dependent and necroptosis independent. What happens with this two type of cell death in your model and is possible that farnesiferol B be able to modulated this pathways in its beneficial effect in renal damage? The identification of a pathway that contribute to AKI persistence such as regulated necrosis or ferroptosis is an expanding research field with important implications for acute kidney injury (AKI) and an analysis of the component of this signalling pathways improve the quality of the study (necroptosis: RIPK1, RIPK3, p-MLKL or ferroptosis: IL-33; GPX4; lipid perox- idation and GSH levels ). The AKI also is characterized by a phase of proliferation after injury to restore the tissue. What is the role of Farnesiferol B in the tubular cells proliferation? Include a staining of PCNA in the renal tissue to show the effect on the restore phase. In the figure 3 is necessary that the authors characterized other inflammatory populations in the kidney such as T lymphocytes (CD3+), CD4+ lymphocytes and neutrophils because there are no previous studies of the effect of this compound in renal damage. Also in necessary a quantification of the positive staining of each type of inflammatory cells. The authors only show the inflammatory cytokines expression by mRNA in the kidney, what happens with the protein levels of MCP1, TNF-alpha and LBT4 in the kidney (It is possible that determined by ELISA from the total protein renal extracts). In the figure 4 the authors only show the effect of Farnesiferol B in the modulation of NFkB signalling in a macrophage cells line assessing nuclear p65 protein levels and transcriptional activity. Is necessary to see the local effect of the treatment in the damaged tissue. Have Farnesiferol B the same effect on NFkb signalling pathway in tubular epithelial cell such as HK2 cell line and in the kidney protein extracts?. Previous studies have described that tubule-epithelial cells expressed TGR5 (Diagn Pathol. 2018 Apr 2;13(1):22). It is possible that farnesiferol B reduce the expression of proinflammatory makers and diminish the NFkb signalling pathway activation in tubular cells? Is possible that that Farnesiferol B modulates the levels of antioxidant molecules such as HO-1, NRF2 and NQOD1? In necessary to improve the discussion of the manuscript. Some of the results such as modulation of oxidative stress is not include in the text and other is not discussed deeply.MINOR POINTS
Why do the authors select the time (2h) and dose ( 20 uM) of Farnesiferol B to their studies in vitro? In the manuscript do not show time-dose experiments. In lane 54 there is a mistake “sesquiterpenethat”, this mistake have to be substituted by “sesquiterpene that”. In necessary to add a picture that summarize the signalling pathway of Farnesiferol B in the kidney.Reviewer 2 Report
The topic of the manuscript by Zhang and colleagues is undoubtedly relevant, ischemia reperfusion injury is a major issue in organ transplantation. The work reports the effects of Farnesiferol B advances in renal ischemia reperfusion injury, suggesting the molecular pathway which mediates the control of these effects.
From the experimental standpoint the research is properly designed and the work has been adequately carried on.
The paper is well written, although there are some careless mistakes that has to be fixed.
My only concern is related to the introduction, which is too short and extremely superficial. I think that this section should be improved in terms of amount of information and scientific soundness.